# Environmental and Circadian Regulation Combine to Shape the Rhythmic Selenoproteome

**DOI:** 10.3390/cells11030340

**Published:** 2022-01-20

**Authors:** Holly Kay, Harry Taylor, Gerben van Ooijen

**Affiliations:** School of Biological Sciences, University of Edinburgh, Max Born Crescent, Edinburgh EH9 3BF, UK; Holly.Kay@ed.ac.uk (H.K.); harryhmtaylor@gmail.com (H.T.)

**Keywords:** selenocysteine, circadian clock, selenoproteome, selenium, cellular rhythms

## Abstract

The circadian clock orchestrates an organism’s endogenous processes with environmental 24 h cycles. Redox homeostasis and the circadian clock regulate one another to negate the potential effects of our planet’s light/dark cycle on the generation of reactive oxygen species (ROS) and attain homeostasis. Selenoproteins are an important class of redox-related enzymes that have a selenocysteine residue in the active site. This study reports functional understanding of how environmental and endogenous circadian rhythms integrate to shape the selenoproteome in a model eukaryotic cell. We mined quantitative proteomic data for the 24 selenoproteins of the picoeukaryote *Ostreococcus tauri* across time series, under environmentally rhythmic entrained conditions of light/dark (LD) cycles, compared to constant circadian conditions of constant light (LL). We found an overrepresentation of selenoproteins among rhythmic proteins under LL, but an underrepresentation under LD conditions. Rhythmic selenoproteins under LL that reach peak abundance later in the day showed a greater relative amplitude of oscillations than those that peak early in the day. Under LD, amplitude did not correlate with peak phase; however, we identified high-amplitude selenium uptake rhythms under LD but not LL conditions. Selenium deprivation induced strong qualitative defects in clock gene expression under LD but not LL conditions. Overall, the clear conclusion is that the circadian and environmental cycles exert differential effects on the selenoproteome, and that the combination of the two enables homeostasis. Selenoproteins may therefore play an important role in the cellular response to reactive oxygen species that form as a consequence of the transitions between light and dark.

## 1. Introduction

Circadian clocks have evolved in all kingdoms of life as an adaptation to the rhythmic environment on planet Earth. An organism’s circadian clock enables the anticipation of predictable daily changes in the environment [1]. When an organism is subjected to a rhythmic environment, such as light/dark cycles, the circadian clock orchestrates processes such as metabolism and photosynthesis to synchronise to the external cycles. Circadian regulation exists at all levels of cellular organisation, including the rhythmic expression of up to one-third of the transcriptome [2,3]. On top of circadian regulation, strong direct responses exist to all rhythmic environmental parameters, but in particular to light/dark cycles. Direct responses to environmental cycles can be disentangled from circadian effects by studies performed under constant environmental conditions.

In a eukaryotic cell, levels of reactive oxygen species (ROS) oscillate over the 24 h cycle, and are influenced by the circadian clock as well as by environmental light/dark cycles [4,5,6,7,8]. Oscillations in the cell redox state are created either from ROS production by rhythmic cellular metabolism (via the mitochondria or NADPH oxidases, or photosynthesis in the green lineage), or via the rhythmic expression of antioxidant proteins within the cell [5]. Powerful oxidising agents, ROS are small, short-lived molecules, involved in the regulation of multiple cell functions and pathways, and include superoxide (O_2_^−^) and hydrogen peroxide (H_2_O_2_). Excessive ROS accumulation can result in oxidative damage and increase cell toxicity, eventually leading to cell death. However, balanced fluctuations of reduction–oxidation (redox) homeostasis have important signalling functions [5,9], implying a tight regulation of redox over the 24 h cycle to compensate for, and integrate, the rhythmic environment.

One group of proteins predominantly involved in redox homeostasis are selenoproteins. First identified by Thressa Stadtman in 1973 [10], selenoproteins are proteins containing a selenocysteine residue. Known as the 21st amino acid, selenocysteine (Sec, or U) is not directly encoded for in the standard genetic code; instead, it is encoded by the opal stop codon UGA through translational recoding that relies on a highly specific *cis*-acting stem–loop structure in the 3′ UTR of the transcript (the selenocysteine insertion sequence; SECIS [11,12,13]). Many selenoproteins function as redox-related enzymes, and selenoenzymes confer a higher catalytic activity than enzymes with a normal cysteine in the active site: the selenol group is far more reactive than the thiol group in a cysteine amino acid, making selenoproteins highly physiologically active [11]. Selenoproteins play critical roles in regulating the cellular redox state and the prevention and repair of ROS-induced damage to cellular components [12,14]. Although selenoproteins are found within bacteria, archaea, and eukaryotes, many organisms have lost the machinery to utilise selenium to create selenocysteine [14], and have cysteine residues in the corresponding positions of enzymes.

Species of phytoplankton perform nearly half of the world’s photosynthesis, and play a critical role as primary producers in global food webs [15]. *Ostreococcus tauri* is one of the major species of marine phytoplankton—a eukaryotic green alga with a small, minimal genome of 12.6 million base pairs, made up of 20 haploid chromosomes [16]. The most recent gene models for this organism reveal that of the 7700 genes contained in its genome, 24 encode for selenoproteins [17]. *Ostreococcus* has a simple, plant-like transcriptional circadian clock [18], which is regulated at the post-translational and metabolic levels by processes shared between eukaryotes [19,20,21,22]. We previously published a detailed proteomic time series in *Ostreococcus tauri* under natural cycles of light and dark (LD; 12 h/12 h cycles of light/dark), as well as under constant light conditions (LL) [23]; that dataset covered all of the 24 *Ostreococcus* selenoproteins, allowing an unprecedented insight into environmental and circadian regulation of the selenoproteome in a eukaryotic model cell.

In this study, we assessed the rhythmicity of selenoproteins under environmental 24 h cycles of LD versus under circadian conditions of LL. We also investigated the effects of the light/dark cycle and the circadian clock on cellular selenium uptake and, reciprocally, the effect of selenium deprivation on the circadian clock. Our combined results point to complex interactions between endogenous circadian regulation and environmental light/dark cycles that together shape the temporal selenoproteome.

## 2. Materials and Methods

Proteomics data were mined from our publicly available previous study [23], and results from samples were collected every 3.5 h across one cycle of 12 h light/12 h dark (LD, or entrained conditions) and three cycles under constant light conditions (LL, or circadian conditions). Cells were grown as documented previously.

Selenoproteins were defined as proteins containing a selenocysteine (U). Selenoproteins were named using the gene function and protein domain information available on ORCAE [24], as well as the functions of homologous proteins in other model species. These protein homologs were determined by Domain Enhanced Lookup Time Accelerated Basic Local Alignment Search Tool (DELTA-BLAST) [25], using the amino acid sequence of the *Ostreococcus* protein. The DELTA-BLAST searches were initially limited to homologs in *Homo sapiens*, *Mus musculus*, and *Drosophila melanogaster*. If there were no closely related homologous proteins in these model species, the search was widened and the three most closely related homologs—which were generally algal species—were retained. To assist in determining the functions of these selenoproteins and their homologs, Gene Ontology (GO) codes associated with *Ostreococcus* selenoproteins and related homologs were taken from UniProt, and GO definitions were taken from QuickGO [26]. To determine whether any selenoproteins were targeted to the organelles within the alga, TargetP-2.0 was used to identify the sequences found before the N-termini of the proteins [27]. Signal peptides were denoted as thylakoid luminal transit peptide (luTP), chloroplast transit peptide (cTP), mitochondrial transit peptide (mTP), or secretory pathway (SP) [23].

The rhythmicity parameters of proteins in both LD and LL were calculated using the eJTK_cycle algorithm in BioDare2 [28], with linear detrending for LL data and without detrending for LD data, as fully explained in [23]. Phase and relative amplitude were calculated using MFOURFIT and the LL proteome period was calculated by MESA, as reported previously [23]. A full rationale for the analysis methods and evidence for the accuracy of the rhythmicity parameters for both the single LD cycle and 3 LL cycles were reported previously [23].

ICP-MS analyses for the selenium isotopes ^78^Se and ^82^Se were performed as previously described [29]. Luminescence data were collected on a TriStar 2 plate reader (Berthold Technologies, Bad Wildbad, Germany) under LD or LL conditions, using 2 µmol/m^2^/s of blue light (Moonlight Blue filter, Lee Lighting). The CCA1-LUC line is a translational fusion of the CCA1 protein with firefly luciferase, driven from the *CCA1* promoter (*pCCA1*::CCA1-LUC), and this line was described elsewhere [18].

Statistics and graphs from the transcriptome and proteome data were calculated and plotted using GraphPad Prism Version 9.1.0. Unless otherwise stated, the statistical tests performed were nonparametric two-tailed Mann–Whitney U tests, with their significance indicated (ns = *p* > 0.05; * = *p* ≤ 0.05; ** = *p* ≤ 0.01; *** = *p* ≤ 0.001; **** = *p* ≤ 0.0001).

## 3. Results

### 3.1. The Selenoproteome over Diurnal and Circadian Cycles

The nuclear genome of the green alga *Ostreococcus tauri* encodes 24 selenoproteins [17], while no predicted selenoproteins exist in the chloroplast or mitochondrial genomes [30]. We analysed the presence of predicted transit peptides at the N-termini of the selenoprotein sequences, and found that while a small number of proteins are predicted to be targeted to the mitochondrion (mitochondrial transit peptide; mTP) or secretory pathway (SP), none have chloroplast transit peptides (Figure 1a). The vast majority of selenoproteins have no previously described transit peptide (None); therefore, it is likely that most selenoproteins are localised in the cytoplasm, and all of them are translated there.

### 3.2. Differential Regulation of Selenocysteine Usage by Environmental and Circadian Rhythms

To gauge the overall level of rhythmic regulation of the selenoproteome, we compared the rhythmicity of selenoproteins versus the overall proteome. Rhythmicity of all protein abundance profiles was assessed using the eJTK algorithm [31,32], as described previously [23]. Abundance profiles over the time series can be found for all selenoproteins in Appendix A. Compared to the overall *Ostreococcus* proteome, a lower proportion of selenoproteins were rhythmic under LD conditions (Figure 2a, Table 1). Surprisingly, the opposite was true under circadian conditions of constant light—a higher proportion of the selenoproteome was rhythmic compared to the overall proteome (Figure 2a).

We next assessed the prevalence of the selenocysteine amino acid (Sec, or U) among rhythmic and arrhythmic proteins. As expected from the observed proportion of rhythmic selenoproteins, there was a significantly increased incidence of selenocysteines in rhythmic compared to arrhythmic proteins under LL conditions, while under LD conditions an opposite trend was observed that did not reach statistical significance (Figure 2b). The opposite trend for selenocysteine usage in rhythmic versus arrhythmic proteins under LD and LL conditions is remarkable, as overall the amino acid usage of the 20 standard amino acids is fairly consistent between LD and LL (Appendix A).

### 3.3. Phase Coordination of the Rhythmic Selenoproteome

We then examined the circadian characteristics of the selenoproteins more closely. In accordance with the overall proteome [23], there was no difference in selenoprotein abundance between LD and LL conditions (Figure 3a); there was also no difference in relative amplitude between rhythmic selenoproteins (Figure 3b). Interestingly, this is a marked deviation from the trend in the overall proteome; generally, proteins oscillated with a far higher relative amplitude under LD cycles than under LL conditions [23]. Therefore, the light/dark cycle appears to exert less of an effect on the extent of selenoprotein rhythms than what would be expected based on the general proteome trends, while the circadian clock has a greater influence on selenoprotein abundance.

Only three selenoproteins were rhythmic under both conditions (Figure 3c), indicating that, similar to the overall proteome [23], selenoprotein rhythmicity is very different between LD and LL conditions. One of these three was a glutathione peroxidase (ostta08g03450), and two of them were thioredoxin-like proteins (selenoprotein H; ostta02g02950, and selenoprotein F; ostta10g01410) [33,34]. Interestingly, when time of peak phase for rhythmic selenoproteins was plotted against the amplitude of their rhythms, we observed that two of these proteins (ostta08g03450 and ostta10g01410) peak together, but at very different peak phases between the two conditions: near ZT6 in LD and CT16 in LL (Figure 3d). The third protein (ostta02g02950) followed this trend, but in the opposite direction to the other two. To indicate this effect more clearly, the protein abundance traces of the glutathione peroxidase are shown in Figure 3e, where the daytime peak under LD conditions and the subjective night peak under LL are evidently different.

When comparing the phase versus amplitude of all rhythmic selenoproteins under LD conditions, we observed no correlation between these two parameters (Figure 3d; r^2^ = 0.06); relative amplitude was consistent across rhythmic proteins regardless of phase. However, under constant light conditions there was a clear trend towards increased amplitude of oscillations in selenoproteins that peak at a later phase during the 24 h cycle (r^2^ = 0.79). Combined, the data in Figure 2 and Figure 3 suggest that rhythmicity within the selenoproteome is regulated by a complex integration of environmental light/dark cycles and endogenous rhythms.

### 3.4. Diurnal Transcript Abundance Rhythms Do Not Dictate Selenoprotein Abundance

To test whether selenoprotein rhythmicity results from rhythmicity of the encoding transcripts, we compared selenoproteome profiles under LD cycles to publicly available transcriptomic data [35], in which we found 15 of the 24 selenoproteins (Appendix A). However, we did not observe a simple relationship between the timing of rhythmic transcripts and selenoproteins; for the six selenoproteins that were rhythmic at both the transcript and protein levels, only one protein peak phase lay between the 2–6 h after the transcript phase that would indicate a correlation [36,37], and the other five did not (Figure 4a, shaded box). An example is provided in Figure 4b, where the protein and transcript phase under LD cycles are close to antiphasic (ostta12g02030; disulphide isomerase)—transcript abundance shoots up directly following dusk, and remains high until the early morning, but this stark increase in transcript abundance after dusk is not followed by an increase in protein abundance until after dawn. Clearly, protein abundance depends not only on transcript abundance but also on the environmental light/dark cycle.

### 3.5. Cellular Selenium Uptake Is Regulated by the Light/Dark Cycle

Selenoprotein synthesis requires an availability of selenium. To test whether instead of transcript abundance, selenoprotein abundance actually follows on from regulated selenium uptake by cells, we performed elemental composition analysis of cells. Inductively coupled plasma mass spectrometry (ICP-MS) was performed on whole-cell extracts to determine cellular selenium content under LD or LL conditions (Figure 5). Under LD conditions, the abundance of both observable isotopes of selenium was extremely rhythmic, with a sharp peak (nearly threefold increase) at the light-to-dark transition (Figure 5a); however, this uptake peak was almost entirely lost under constant circadian conditions (Figure 5b; note the different y-axis scales between panels a and b), indicating that selenium uptake rhythms are mediated through environmental signalling rather than endogenous circadian regulation. As selenium content was only elevated around dusk under LD conditions, and there was no apparent increased selenoprotein content at that time, we conclude that selenium uptake rhythms do not underlie selenoprotein rhythmicity.

### 3.6. Functional Effects of Selenium Deprivation on Cellular Timekeeping

Finally, as we identified differential effects of light/dark cycles or constant circadian conditions on the abundance of selenoproteins as well as on selenium uptake, we tested the effects of selenium deprivation on circadian clock gene expression under both conditions. Cells were grown with normal concentrations of selenium (1 × 10^−8^ M, provided in the media as selenious acid; H_2_SeO_3_) or at 2 or 4 orders of magnitude lower. Clock gene expression was tested by longitudinal luminescent imaging of the circadian clock protein CCA1 translationally fused to the firefly luciferase enzyme [18]. Under LD conditions, selenium deprivation led to decreased amplitude of CCA1 expression rhythms (Figure 5c). More pronounced, however, selenium deprivation led to perturbed dynamics of clock gene expression around the light/dark transitions—at dusk, and especially at dawn, selenium-deprived cells responded with a large shoulder of CCA1 expression, while the control cells were not greatly affected by the light/dark transitions. Under constant circadian conditions, the lower amplitude in selenium-deprived cultures was consistent with LD, but the traces were qualitatively identical, without major period or phase defects (Figure 5d). Combined, this could imply that selenoproteins are important for the anticipation of—and compensation against—the effects of the light/dark cycle on timekeeping.

## 4. Discussion

We observed a limited similarity between rhythmicity of the selenoproteome under environmentally rhythmic conditions and circadian conditions of constant light—the identity of rhythmic selenoproteins and their peak abundance phases were different. However, in the absence of environmental rhythms, it is clear that the circadian clock shapes the selenoproteome over time; selenoproteins are overrepresented among rhythmic proteins under constant light conditions (Figure 2), and we observed a clear association between abundance of selenoproteins and circadian phase (Figure 3). This indicates a coordinated upregulation of these redox-related enzymes throughout the day. Under environmentally rhythmic conditions, on the other hand, the peak abundance of selenoproteins is spread evenly throughout the 24 h cycle, implying that the circadian regulation observed under LL conditions integrates with environmental signalling. This might mean that the circadian regulation that can be observed under constant light conditions exists merely to anticipate and counteract the effects of the environmental rhythms to enable homeostasis. Clearly, the selenoproteome is shaped by environmental cycles in combination with anticipatory effects achieved by circadian timekeeping.

A lack of correlation was observed between selenoprotein rhythmicity and their cognate transcript rhythms. This indicates that the rhythmicity of selenoproteins is defined by post-transcriptional processes. A clear shortcoming of this conclusion is that no transcriptome is available under constant circadian conditions for *Ostreococcus*, so analyses are limited to LD conditions. However, consistent with an overall proteome analysis [23], we conclude that selenoprotein abundance rhythms cannot be assumed from transcript data alone, nor can the rhythmicity of the protein under physiological rhythmic conditions be inferred from rhythmicity under constant circadian conditions.

The translation of selenoproteins requires selenium ions, which are lowly but sufficiently abundant in the sea water in which *Ostreococcus* grows. We pursued the hypothesis that rhythmically regulated selenium uptake might be the cause of increased prevalence of selenoproteins among clock-regulated proteins; however, the opposite was true—under circadian conditions we found no high-amplitude rhythms of cellular selenium content, whereas a clear diurnal pattern of selenium uptake was observed under light/dark cycles, with a peak directly following dusk. As this peak was largely absent under constant circadian conditions, it must result from a direct effect of darkness. Under light/dark cycles, however, selenoproteins are less likely to be rhythmic compared to the overall proteome, indicating that selenium uptake rhythms do not translate linearly to strong selenoprotein abundance rhythms. However, this result highlights the fact that selenium uptake is one of the effects of environmental rhythms that might integrate with circadian effects to influence the selenoproteome. Similarly, we found that selenium deprivation strongly affects the dynamics of circadian clock protein CCA1 under diurnal cycles, but not circadian conditions. When cells are grown in the absence of selenium, translation of selenoproteins terminates at the UGA codon, resulting in a truncated, non-functional enzyme [13]. It is therefore reasonable to assume that selenium depletion leads to lower levels of selenoproteins. As selenoproteins are a major class of redox homeostasis enzymes, and light/dark transitions are known to cause fluctuations in redox, it is tempting to speculate that selenoprotein activity buffers circadian gene expression against the effects of the light/dark cycle on the cellular redox state. As with our earlier results, this would be consistent with the notion that circadian regulation of the selenoproteome enables homeostasis under physiologically normal conditions.

In eukaryotes, only the genomes of mammals, small animals, plasmodia, and green algae encode confirmed selenoproteins [14]. This association with primary producers in marine/aquatic environments, where selenium is present, and in animals that are higher in the food web has led to the idea that other species—such as land plants—lost selenocysteine by necessity, as there is no or too little selenium available in their environment [14]. Given the huge importance of *Ostreococcus tauri* for carbon capture and marine food webs as a major species of marine phytoplankton, it is relevant to understand how their biochemistry is controlled by endogenous and environmental factors in the face of a changing environment. In addition to this direct relevance, *Ostreococcus* is now a bona fide model cell for the study of eukaryotic circadian rhythms that shares metabolic rhythms with human cells [22,29,38]. Therefore, this investigation can also directly inform studies on circadian regulation of the human or other vertebrate selenoproteomes, where a complete proteomic dataset would be harder to obtain.

## Figures and Tables

**Figure 1 cells-11-00340-f001:**
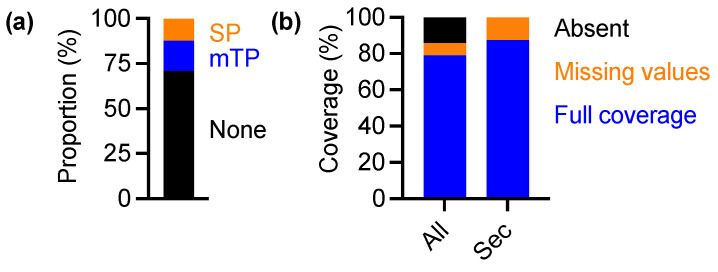
Proteome coverage of the selenoproteome: (**a**) Proportion of selenoproteome predicted to have a transit peptide for the secretory pathway (SP), mitochondrion (mTP), or no transit peptide (None). (**b**) Coverage of proteins containing a selenocysteine (Sec) versus the whole proteome (All) in [23]. ‘Absent’ refers to proteins that were not quantified at any of the timepoints in either the LD or LL time series. ‘Missing values’ refers to proteins that were quantified in some but not all data points of the time series, while ‘Full coverage’ refers to those proteins that were reliably quantified at every timepoint throughout the LD and LL time series. We previously published detailed proteomic time series in *Ostreococcus tauri* under natural cycles of light and dark (LD; 12 h/12 h cycles of light/dark), as well as under constant light conditions (LL), sampling at 3.5 h intervals [23]. In that study, we obtained 85% coverage of the theoretical proteome and, interestingly, all 24 selenoproteins were detected in the dataset (Figure 1b and Appendix A). A total of 21 out of 24 selenoproteins had full coverage over the LD and LL time series (blue in Figure 1b), while 3 had missing data points (orange). Based on this exceptional coverage, we conclude that our dataset provides an unprecedented insight into rhythmic regulation of the selenoproteome.

**Figure 2 cells-11-00340-f002:**
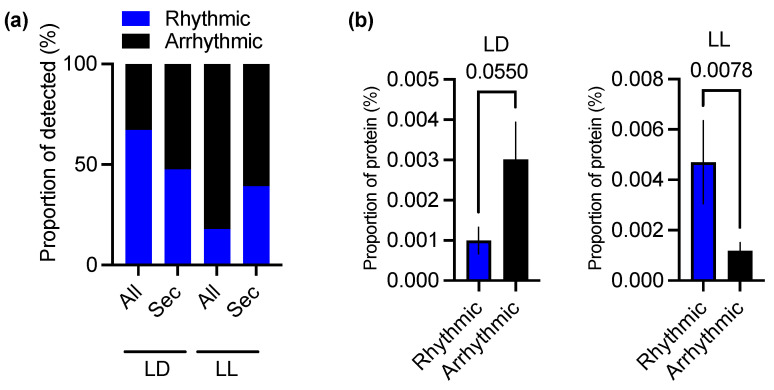
Rhythmicity of the selenoproteome: (**a**) Proportion of rhythmic versus arrhythmic proteins under LD or LL in the overall dataset reported in [23], or those proteins containing a selenocysteine (Sec). (**b**) Amino acid usage for selenocysteine, expressed as the average proportion of a protein amongst rhythmic or arrhythmic proteins under LD (left) or LL (right) conditions. Statistics are unpaired nonparametric tests (Mann–Whitney).

**Figure 3 cells-11-00340-f003:**
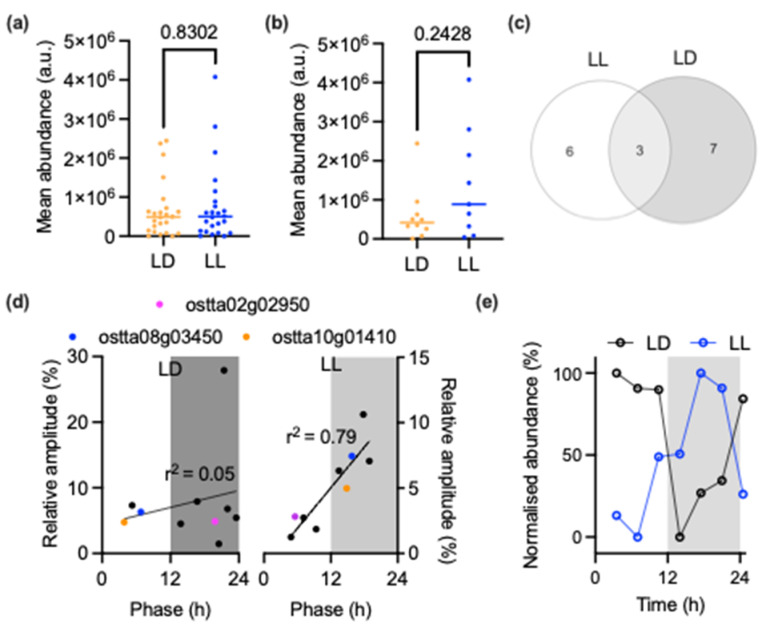
Rhythmicity parameters of selenoproteins: (**a**) Mean abundance of all selenoproteins under LD versus LL conditions. (**b**) Mean abundance of selenoproteins classed as rhythmic under LD versus LL conditions. Statistics in (**a**,**b**) are unpaired nonparametric tests (Mann–Whitney); data are reported in arbitrary units (a.u.). (**c**) Overlap between rhythmic selenoproteins under LD versus LL conditions. (**d**) Peak abundance phase versus relative amplitude of rhythmic selenoproteins under LD (left) or LL (right) conditions. Coloured data points represent proteins that are arrhythmic under both conditions. Lines are linear regression with annotated r^2^ values. (**e**) Protein abundance across LD versus LL for ostta08g03450. The LD data reflect the single LD cycle, while the LL data are mean values across the three LL cycles in [23].

**Figure 4 cells-11-00340-f004:**
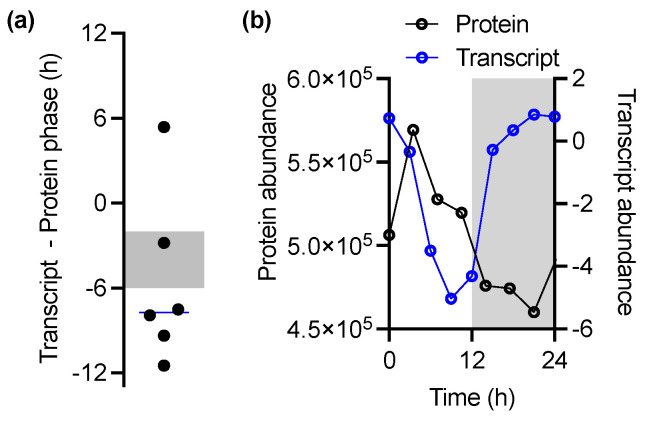
No correlation between protein and transcript peak abundance phase: (**a**) One-for-one phase relationship between rhythmic selenoproteins and their cognate transcripts, expressed as the peak abundance phase of the transcript minus the peak abundance phase of the protein. Values that correspond to a protein that peaks in a 2–6 h window after its transcript would map to the shaded area. (**b**) Protein versus mean transcript abundance across the LD cycle for ostta12g02030.

**Figure 5 cells-11-00340-f005:**
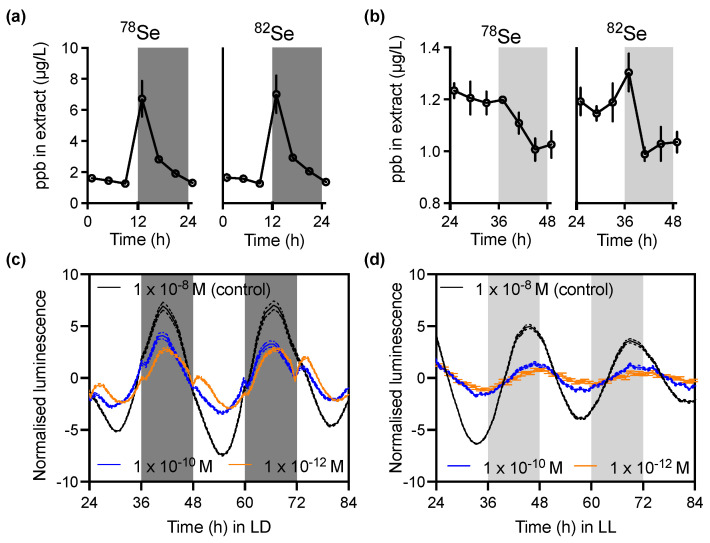
Interaction of selenium with the environmental and circadian cycles: (**a**,**b**) Cellular content of selenium, analysed by inductively coupled plasma mass spectrometry. The two detectable stable isotopes of selenium are ^78^Se (left) and ^82^Se (right), and these were measured under LD (**a**) or LL (**b**) conditions. Mean ± SEM; *n* = 3. (**c**,**d**) Luminescent imaging of the CCA1-LUC circadian clock marker under normal physiological concentrations of extracellular selenium (control) versus depleted levels 2 or 4 orders of magnitude lower. Cells were imaged over multiple cycles of 12 h/12 h light/dark (LD, (**c**)) or constant light (LL, (**d**)). Mean ± SEM; *n* = 32.

**Table 1 cells-11-00340-t001:** The Ostreococcus selenoproteins and rhythmicity parameters under light/dark cycles (LD) or constant light (LL): Provided are the *p*-value for rhythmicity, phase (in hours; h), relative amplitude (Rel. Amp., in percentage), and mean abundance (Mean Ab., in arbitrary units; a.u.) of the selenoproteins, as detected in our previous proteomics study [23]. ND = not done, meaning the analysis was not possible due to too many missing values in the dataset.

Identifier	Description	LD				LL				
		*p*-Value	Phase (h)	Rel. Amp. (%)	Mean Ab. (a.u.)	*p*-Value	Period (h)	Phase (h)	Rel. Amp. (%)	Mean Ab. (a.u.)
ostta01g05530	Thioredoxin-fold protein	0.530	16.3	0.8	633,478	0.000	21.7	7.1	2.7	647,647
ostta10g01410	Selenoprotein F	0.014	3.9	4.8	76,668	0.007	23.1	14.9	5.0	84,005
ostta01g00700	Disulphide isomerase 1	0.182	5.2	2.5	2,371,875	0.008	22.5	13.5	6.3	2,144,215
ostta09g01390	Selenoprotein U	0.061	20.7	2.2	1,505,570	0.012	18.6	4.8	1.2	1,433,558
ostta02g02950	Selenoprotein H	0.018	19.8	4.9	353,763	0.016	25.6	5.6	2.8	322,590
ostta08g03450	Glutathione peroxidase A	0.000	6.8	6.3	2,441,906	0.027	21.6	15.8	7.4	4,076,098
ostta09g00190	Peroxiredoxin	0.182	8.6	3.2	2,093,776	0.027	24.1	18.9	7.1	2,806,390
ostta14g01560	Selenoprotein K	ND	ND	ND	41,402	0.031	25.7	9.4	1.9	41,354
ostta01g06300	Selenoprotein W	0.123	9.3	5.9	574,635	0.039	24.2	17.8	10.6	889,748
ostta09g00530	Glutathione peroxidase C	0.108	21.5	1.2	395,005	0.063	21.6	17.1	4.7	514,482
ostta09g01720	Methionine sulphoxide reductase A	ND	ND	ND	1577	0.063	22.0	17.3	9.6	2771
ostta12g02030	Disulphide isomerase 2	0.000	5.3	7.3	503,473	0.102	22.2	5.9	3.8	377,407
ostta02g02735	Glutathione peroxidase E	0.000	16.7	7.9	628,421	0.102	22.1	10.1	3.6	587,297
ostta17g00710	SAM-dependant methyltransferase	0.037	20.5	1.4	485,420	0.125	21.3	4.8	2.1	487,887
ostta01g04220	Thioredoxin reductase	0.008	13.8	4.5	951,929	0.219	24.4	1.6	2.5	1,150,422
ostta08g03600	Unknown	0.000	22.0	6.8	258,959	0.281	25.8	3.7	3.2	262,933
ostta18g01790	Selenoprotein O	0.016	23.6	5.4	328,866	0.281	23.3	11.1	0.6	379,395
ostta05g01540	Glutathione peroxidase B	0.197	7.9	2.6	719,605	0.406	21.9	9.3	1.2	765,471
ostta10g02090	Membrane selenoprotein	0.009	21.4	27.9	6724	0.656	27.6	2.7	10.0	6528
ostta07g00300	Glutathione peroxidase D	0.061	16.5	1.5	552,907	0.688	34.2	23.3	0.6	612,140
ostta13g00280	Disulphide isomerase 3	0.669	13.9	0.7	604,624	0.750	23.4	8.4	2.2	601,203
ostta10g00035	Selenoprotein S	0.106	22.5	10.7	144,751	0.781	21.8	16.4	4.0	137,022
ostta03g04910	Selenoprotein T	0.106	16.5	1.8	113,777	0.844	17.8	3.8	1.2	114,652
ostta04g01370	Selenoprotein M	ND	ND	ND	68,795	ND	ND	ND	ND	77,943

## Data Availability

The mass spectrometry data were mined from publicly available data at the ProteomeXchange Consortium, with the dataset identifier PXD025009.

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
