# Peer review of "Environmental and Circadian Regulation Combine to Shape the Rhythmic Selenoproteome"

_cells, 2022, doi:10.3390/cells11030340_

Round 1

Reviewer 1 Report

The manuscript entitled “Environmental and circadian regulation combine to shape the rhythmic selenoproteome” described that seleno proteins are over-represented among rhythmically abundant proteins under LL, but under-represented under LD conditions in the picoeukaryote Ostreococcus tauri. The circadian and environmental cycle exert differential effects on the selenoproteome that combines to allow homeostasis, while the circadian clock has a greater influence on selenoprotein abundance. The rhythmicity within the selenoproteome is regulated by a complex integration of environmental light-dark cycles and endogenous rhythms. Moreover, they found that selenium uptake rhythms do not underlie selenoprotein rhythmicity. The logical structure of this manuscript is compact, but there are some problems that need to be corrected.

  1. The content of Figure 1b is not described and quoted in the text. The explanation of Figure 1b in the legend is not clear, so it is difficult for readers to understand the meaning of this figure.

The content of “the mitochondrion or secretory pathway, none have chloroplasttransit peptides” in the line 130 should add the abbreviation of mitochondrion (mTP), secretory pathway (SP) and none have chloroplasttransit peptides (None).

“In that study we obtained 85% coverage of the theoretical proteome, and interestingly all 24 selenoproteins were detected in the dataset (Figure 1a and Table S1).”, which may be the context of Figure 1b.

  1. “Surprisingly, the opposite was true under circadian conditions of constant light: a higher proportion of the selenoproteome was rhythmic compared to the overall proteome.”, which should add the Figure 2a.
  2. The full name of unit “a.u.” in figure 3 should be added in the figure legends. In line 183, “phases between the two conditions: near ZT6 in LD and CT16 in LL (Figure 3d, arrows)”, but there is no arrow in Figure 3d.
  3. “The third protein follows this trend but in the opposite direction to the other two.”, while you did not specify which is the third protein in the article.
  4. The “CCA1-LUC” in figure 5c legends should be “CCA1::LUC or P CCA1:LUC”.
  5. The format of some reference titles are not uniform.
  6. Ostreococcus tauri” in the reference titles were not italicized.
  7. Some reference names were not in right format, such as Plos one, Trends in biochemical sciences.
  8. In full text, the 24h should be 24 h.

Author Response

Reviewer 1

The authors would like to thank both reviewers for their comments, which have certainly improved the manuscript. You have found some inconsistencies and mistakes that we have been able to rectify in full thanks to your critical eye. Below, please find our replies to your individual comments.

The content of Figure 1b is not described and quoted in the text. The explanation of Figure 1b in the legend is not clear, so it is difficult for readers to understand the meaning of this figure.

“In that study we obtained 85% coverage of the theoretical proteome, and interestingly all 24 selenoproteins were detected in the dataset (Figure 1a and Table S1).”, which may be the context of Figure 1b.

You are right; that reference should have been to figure 1b, not 1a. We have amended this mistake, added more clarification and trust it is now clear. We have also included a more stand-alone description of Figure 1b in the figure legend.

The content of “the mitochondrion or secretory pathway, none have chloroplasttransit peptides” in the line 130 should add the abbreviation of mitochondrion (mTP), secretory pathway (SP) and none have chloroplasttransit peptides (None).

We have amended this.

“Surprisingly, the opposite was true under circadian conditions of constant light: a higher proportion of the selenoproteome was rhythmic compared to the overall proteome.”, which should add the Figure 2a.

We have added a reference to Figure 2a as suggested to line 165.

The full name of unit “a.u.” in figure 3 should be added in the figure legends.

In line 183, “phases between the two conditions: near ZT6 in LD and CT16 in LL (Figure 3d, arrows)”, but there is no arrow in Figure 3d.

The legend to figure 3 has been amended as suggested.

“The third protein follows this trend but in the opposite direction to the other two.”, while you did not specify which is the third protein in the article.

We have added specification of the third protein.

The “CCA1-LUC” in figure 5c legends should be “CCA1::LUC or P CCA1:LUC”.

We have added details of the line to the methods: " The CCA1-LUC line is a translational fusion of the CCA1 protein with firefly luciferase, driven from the CCA1 promoter (pCCA1::CCA1-LUC), and this line was described elsewhere (ref Corellou et al.)" and have also mentioned it is a translational fusion in the corresponding results section.

The format of some reference titles are not uniform.“Ostreococcus tauri” in the reference titles were not italicized. Some reference names were not in right format, such as Plos one, Trends in biochemical sciences.

We have amended the formatting problems highlighted here.

In full text, the 24h should be 24 h.

This has been changed throughout.

Reviewer 2 Report

The manuscript describes an extensive analysis of the Ot selenoproteome, following the recently published article about the circadian regulation of the Ot proteome from the same group. The data analysis is quite well performed using the appropriate methods, offering interesting new insights on the Ot selenoproteome. However, there is a need for a more accurate description of the findings. Here are some points that could be improved in the manuscript.

Line 9: Besides the negation of the detrimental effects of redox imbalance, the regulation of the redox state is also important to attain homeostasis, and it can also feedback regulate the clock. Please rephrase the sentence to include these notions. I found the abstract a bit wordy and convoluted, much more than the introduction and the subsequent parts of the text. Simplification would be welcome to provide a clear-cut presentation of the findings.

Line 15: suggestion: “light-dark (LD) cycles”

Line 20: please make clear what are the entrained and circadian conditions. I believe that you mean LD and LL, respectively.

Line 31: there are recent reviews/references about the circadian clocks in plants

Line 43: “…the rhythmic expression of antioxidant proteins”

Line 45: add the ROS molecular formula (e.g.: H2O2, O2-)

Line 56: italicize “cis”

Line 69: citation is inconsistent with the manuscript. Please check the references throughout the text.

Line 80: I suggest using LD for light-dark cycles and LL for continuous/constant light

Line 91: reference 24 seems unnecessary, mainly because reference 23 cites it already, to describe the growth conditions

line 114: Please indicate the statistics and the significance applied to the quantification of Se isotopes and the luminescence from CCA1:LUC. The next paragraph states only the methods for the data mining analyses. Also, I think that the luminescence data is from the LUC driven by the CCA1 promoter. Please make this clearer.

Line 131: the selenoproteins have no “any previously known or described” transit peptide. The in-silico prediction does not exclude the presence of transit peptides that are currently unknown.

Line 132: the authors say that the selenoproteins are active in the cytoplasm. What is the rationale to state that they are active and not inactive?

Table 1 needs reformatting to accommodate the information. It is hard to keep track of the rows you’re in. The table also needs more information: what “Ab” means, absorbance? A.U. are arbitrary units? Are the LD cycles equinoctial? Or long-days or short days? What Rel. Amp. stands for? Relative Amplitude?

Figure 1: please specify clearly what means “full coverage”, “missing values”, etc., in the legend. The figure should stand on its own with the legend.

Supp. Figure 1: please provide evidence (literature/demonstration) that a single cycle (a single day) in light-dark cycles is enough to robustly capture the rhythmicity of gene expression. The same goes for the other figures that rely on this parameter.

Figure 3d: is it accurate to compare LD vs LL since both have a different number of data points collected (1 cycle for LD and 3 cycles for LL)? Is the conclusion merely an artifact due to the reduced number of data collecting points for LD cycles (which may not capture the oscillatory trend correctly through several days) and the failure to capture the real oscillation of the protein accumulation? Please discuss this limitation in the text and adjust the conclusions accordingly.

Line 202: Is there evidence that suggests correlation between transcript and protein abundance when their peaks are 2-6 hours apart? Why the antiphasic peaks of the gene chosen for the analysis showed figure 4 aren’t indicating that they are correlated? The transcript peak is around the end of the night and protein peak occurs in the morning? Could we say they are uncorrelated?

Figure 5: please explain how the uptake of Se78 and Se82 are so similar in the cells if the natural abundance of Se78 (natural abundance approximately 25%) and Se82 (natural abundance around 8%) are so different. For this question I wish to see the original data and the detailed methods of cell growth and Se quantification showed in Figure 5a.

References: Please check the references for not-italicized species names in the article titles. This is a common issue in reference managers. Thus, I suggest adjusting it manually.

Author Response

Reviewer 2

The authors would like to thank both reviewers for their comments, which have certainly improved the manuscript. You have found some inconsistencies and mistakes that we have been able to rectify in full thanks to your critical eye. Below, please find our replies to your individual comments.

 Line 9: Besides the negation of the detrimental effects of redox imbalance, the regulation of the redox state is also important to attain homeostasis, and it can also feedback regulate the clock. Please rephrase the sentence to include these notions. I found the abstract a bit wordy and convoluted, much more than the introduction and the subsequent parts of the text. Simplification would be welcome to provide a clear-cut presentation of the findings.

We agree, and have reworded this to reflect your comment, and made some improvements to the narrative of the abstract, mostly by using fewer specialist terms. We trust it is more accurate and more accessible now.

Line 15: suggestion: “light-dark (LD) cycles”

Amended.

Line 20: please make clear what are the entrained and circadian conditions. I believe that you mean LD and LL, respectively.

We have added “entrained”, so that it is clear that “entrained” refers to LD. The occurrence in what was line 20 has been changed to read "LD" instead.

Line 31: there are recent reviews/references about the circadian clocks in plants.

We have cited a review that compares the rhythmic systems of all the major taxa, including plants. Unfortunately, there are not many of these available, and I still find this reference (Zhang and Kay, 2010) the most illuminating. Given that plants do not contain selenoproteins, I do believe a general clock review makes more sense than a plant-specific one.

Line 43: “…the rhythmic expression of antioxidant proteins”

Line 45: add the ROS molecular formula (e.g.: H2O2, O2-)

Line 56: italicize “cis”

Line 80: I suggest using LD for light-dark cycles and LL for continuous/constant light

Line 91: reference 24 seems unnecessary, mainly because reference 23 cites it already, to describe the growth conditions

All of the above have been amended as suggested.

Line 114: Please indicate the statistics and the significance applied to the quantification of Se isotopes and the luminescence from CCA1:LUC. The next paragraph states only the methods for the data mining analyses. Also, I think that the luminescence data is from the LUC driven by the CCA1 promoter. Please make this clearer.

We have added detail here, to make it clear that this is a translational fusion, as well as in the corresponding results section. " The CCA1-LUC line is a translational fusion of the CCA1 protein with firefly luciferase, driven from the CCA1 promoter (pCCA1::CCA1-LUC), and this line was described elsewhere". No statistics were applied to either the Se isotope quantification or the luminescence data, since we believe the claims we make in the manuscript require only visual inspection of the data presented in the figures.

Line 131: the selenoproteins have no “any previously known or described” transit peptide. The in-silico prediction does not exclude the presence of transit peptides that are currently unknown.

We have amended as per the suggestion.

Line 132: the authors say that the selenoproteins are active in the cytoplasm. What is the rationale to state that they are active and not inactive?

We have changed “active” to “localised” to avoid confusion.

Table 1 needs reformatting to accommodate the information. It is hard to keep track of the rows you’re in. The table also needs more information: what “Ab” means, absorbance? A.U. are arbitrary units? Are the LD cycles equinoctial? Or long-days or short days? What Rel. Amp. stands for? Relative Amplitude?

We agree the way the table has been integrated into the manuscript is not ideal, but this is something the journal has done and in our understanding, this will be improved by the journal in the eventual published version. We will keep an eye on this during the proofing process. However, we apologise for the lack of legend to this table. We have now included a legend: "Table 1: The Ostreococcus selenoproteins and their rhythmicity parameters under 12:12 light-dark cycles (LD) or constant light (LL), showing the p-value for rhythmicity, phase, relative amplitude (Rel. Amp.) and mean abundance (Mean Ab.) of the selenoproteins as detected in our previous proteomics study".

Figure 1: please specify clearly what means “full coverage”, “missing values”, etc., in the legend. The figure should stand on its own with the legend.

We have added some detail to this legend: " ‘Absent’ refers to proteins that were not quantified at any of the timepoints in either LD or LL timeseries. ‘Missing values’ refers to proteins that were quantified in some but not all data points of the time series, while ‘Full coverage’ refers to those proteins that are reliably quantified at every timepoint throughout the LD and LL time series."

Supp. Figure 1: please provide evidence (literature/demonstration) that a single cycle (a single day) in light-dark cycles is enough to robustly capture the rhythmicity of gene expression. The same goes for the other figures that rely on this parameter.

Figure 3d: is it accurate to compare LD vs LL since both have a different number of data points collected (1 cycle for LD and 3 cycles for LL)? Is the conclusion merely an artefact due to the reduced number of data collecting points for LD cycles (which may not capture the oscillatory trend correctly through several days) and the failure to capture the real oscillation of the protein accumulation? Please discuss this limitation in the text and adjust the conclusions accordingly.

This is indeed a matter that we have addressed in great depth when reporting our initial study last year. In that paper we address this is great detail and show all alterantive possibe analyses methods alongside the one we chose, to show that our analyses are the best available. As this is recently published work, we would refer the reviewer and the audience to the relevant parts of that study: supplementary figure 1 of Kay et al., 2021 explains why under LD multiple cycles would not lead to better insights, and supplemental figure 2 shows heat maps for all rhythmicity analysis algorithms to show why we chose these. In the methods section (line 113) we have added “A full rationale for the analysis methods and evidence for accuracy of rhythmicity parameters for both the single LD cycle and 3 LL cycles was reported previously.”

Line 202: Is there evidence that suggests correlation between transcript and protein abundance when their peaks are 2-6 hours apart? Why the antiphasic peaks of the gene chosen for the analysis showed figure 4 aren’t indicating that they are correlated? The transcript peak is around the end of the night and protein peak occurs in the morning? Could we say they are uncorrelated?

We have added references for the “2-6 hour” transcript-protein relationship (Robles et al., 2014 and Kojima et al., 2011). Of course, the data as presented in 2b shows a line through data points, whereas the analysis in 4a depends on the fit through those data points. In that sense, I can see it is harder to interpret the protein and transcript traces. I do not agree that the transcript peak is around the end of the night; it shoots up at the start of the night and stays quite similar for the full night. That stark increase just after dark is not followed by an increase in protein abundance until 12 hours later in the early morning. So there is no correlation at all during the night. I appreciate there might be good correlation if you only look at the light phase. Therefore, this example shows clearly that transcript abundance is not the sole driver of protein abundance, which depends not only on transcript abundance but strongly on the light-dark cycle too.  The predicted peak phases from model fits through these data would confirm that. However, I see that the reviewer, and therefore our audience too, can get confused by this effect and therefore we have added this statement to the results section: " transcript abundance shoots up directly following dusk and remains high until the early morning, but this stark increase in transcript abundance after dusk is not followed by an increase in protein abundance until after dawn. Clearly, protein abundance depends not only on transcript abundance but also on the environmental light-dark cycle." We trust that the reviewer will accept the current wording.

Figure 5: please explain how the uptake of Se78 and Se82 are so similar in the cells if the natural abundance of Se78 (natural abundance approximately 25%) and Se82 (natural abundance around 8%) are so different. For this question I wish to see the original data and the detailed methods of cell growth and Se quantification showed in Figure 5a.

The reviewer raises a very interesting point, and I'm afraid that I cannot say for certain why the different natural abundance does not translate to a difference in selenium isotopes observed in our samples. The only consideration I have here is that we make up our own seawater, and add selenium as a purchased chemical: I can imagine that the % of isotopes in the product are not identical to the % in the natural environment.  By the same logic, concentrations are defined by a standard concentration range of each ion that is measured in the same run of ICP-MS, so the relative abundance of each isotope in the standard would also affect the observed concentration. The data presented in the figures is the raw data after inference of concentration from the standard curve, and after correction for machine drift over the runs. The method for growing cells and extracting elemental ions is fully identical to our previous study employing ICP-MS (Feeney et al., 2016), as we stated in the methods section.

References: Please check the references for not-italicized species names in the article titles. This is a common issue in reference managers. Thus, I suggest adjusting it manually.

We have amended the issues with the reference list.

Round 2

Reviewer 2 Report

The authors performed the requested changes and presented the data. Congratulations for the publication.